# Nutritional outcomes of therapeutic feeding program and its predictors among undernourished adult HIV positive patients at healthcare facilities of West Guji Zone, Southern Ethiopia: A retrospective cohort study

**Eden Ashenafi** [1] *, **Getahun Beyene Guluma**[2,3], **Dirshaye Argaw**[2], **Habtamu Endashaw Hareru**[2], **Nagasa Eshete Soboksa**[4]

**1** Department of Reproductive Health, College of Health and Medical Science, Dilla University, Dilla, Ethiopia, **2** School of Public Heath, College of Health and Medical Sciences, Dilla University, Dilla, Ethiopia, **3** Adolescent Sexual and Reproductive Health Coordinator, Population Service International Ethiopia, Negele Arsi, Ethiopia, **4** Department of Environmental Health, College of Health and Medical Sciences, Dilla University, Dilla, Ethiopia

\* eden.ashenafi@du.edu.et

## Abstract

### Background

For those living with HIV/AIDS, malnutrition is a significant issue everywhere, but it is particularly prevalent in Sub-Saharan Africa. A nutritional support program is becoming a more and more common strategy to prevent malnutrition in HIV-positive persons. Thus, this study aimed to assess nutritional treatment outcomes and their predictors among adult HIV-positive undernourished individuals in West Guji Zone healthcare facilities.

### Method

A facility-based retrospective cohort study was conducted among 348 randomly selected adult HIV- positive patients in the West Guji Zone healthcare facilities between January 2018 and December 2022. Data were collected using the data extraction tool. Entered into Epi Data version 3.1 and exported to SPSS version 26 for analysis. The Kaplan-Meier survival curve and log-rank test were used to predict the time to recovery and to compare survival curves across categorical variables. A Cox proportional hazard regression model was fitted to identify an independent predictor of the recovery rate. Statistical significance was declared at a p-value of < 0.05.

### Results

In the final analysis 348 undernourished HIV-positive persons were included. Based on preset exit criteria, approximately 198, 56.9% of patients enrolled in the RUTF program were able to recovered, with an incidence of 9.83 (95% CI: 3.12, 13.44) per 100 person-month

**Data Availability Statement:** All relevant data are within the manuscript and its Supporting Information files.

**Funding:** The author(s) received no specific funding for this work.

**Competing interests:** The authors have declared that no competing interests exist.

**Abbreviations:** AIDS, Acquired immune deficiency syndrome; ART, Anti-Retroviral Therapy; BMI, Body Mass Index; CD4, Cluster Differentiation 4; CI, Confidence Interval; DC, Data Collector; FBP, Food by Prescription; FMOH, Federal Ministry of Health; HAART, Highly Active Anti-Retroviral Therapy; HIV, Human Immune Virus; MAM, Moderate Acute Malnutrition; NACS, Nutritional Assessment, Counseling and Support; PEPFAR, Presidents Emergency Plan for AIDS Relief; PI, Principal Investigator; PLWH, People Living with HIV; RUTF, Ready to Use Therapeutic Food; SAM, Severe Acute Malnutrition; SPSS, Statistical Package for Social Science; USAID, United States Agency for International Development; WFP, World Food Program; WHO, World Health Organization.

observations. Being divorced (AHR = 0.21; 95% CI: 0.06, 0.69) and being in the WHO advanced stage (AHR = 0.42; 95% CI: 0.23, 0.79) was a negative predictor. Being in the age range of 18–29 and 30–39 and having a working functional status (AHR = 2; 95% CI: 1.25, 3.23) were positive predictors.

## Conclusion

Nutritional recovery in this study lower than WHO Sphere requirements. Age between 18 and 39 and working functional status were good indicators of nutritional recovery, whereas advanced WHO clinical stage and divorced marital status were negative predictors.

## Introduction

The Human Immunodeficiency Virus (HIV) and Acquired Immune Deficiency Syndrome (AIDS) pose significant global public health challenges, particularly in countries of sub-Saharan Africa. A sizable fraction of the population is impacted by HIV/AIDS, whether directly or indirectly. For those living with HIV/AIDS, malnutrition is a significant issue everywhere, but it is particularly prevalent in Sub-Saharan Africa. This crisis has had a negative impact on social health and well-being, adult productivity, and adult survival. The HIV pandemic alarmingly elevated morbidity and mortality rates, which synergize when combined with malnutrition [1,2].

As of the end of 2020, there were 1.5 million new cases of HIV infection, bringing the total number of persons living with the disease to about 37.7 million (with a range of 30.2 to 45.1 million). Adults make up almost 34.5 million of those. HIV/AIDS has had a significant negative impact on developing countries, particularly in Sub-Saharan Africa, which accounts for 1 in every 25 adult (3.2%) living with HIV and accounting for more than two-thirds of the people living with HIV globally [3,4].

HIV prevalence exceeds 25% among adults in eastern and southern Africa, the regions hardest hit by the epidemic. According to 2019 estimates, Ethiopia is home to 860,000 people living with HIV, with prevalence rate of 0.9% [5].

Nearly 50% of people living with HIV worldwide are malnourished, especially in sub-Saharan Africa, including Ethiopia. Their effects are synergistic and can exacerbate underlying conditions. Both damage the immune system and increase the risk of infection, serious illness, and death [6]. Adequate nutrition is critical to preventing opportunistic infections, maintaining immune status, improving response to ART treatment, and achieving the best quality of life for patients living with HIV. By integrating nutritional interventions into chronic HIV care, patients can delay progression to disease stages of infection, maximize antiretroviral benefit, and improve adherence to treatment regimens [7].

HIV prevalence exceeds 25% among adults in eastern and southern Africa, the regions hardest hit by the epidemic. According to 2019 estimates, Ethiopia is home to 860,000 people living with HIV, with prevalence rate of 0.9% [5].

The management of HIV/AIDS must include complete nutrition support. HIV/AIDS and malnutrition are mutually reinforcing and closely related conditions. Malnutrition is more common in those with HIV, but it also hastens the disease's progression to the most severe stage. They seriously damage the immune system, making people more susceptible to disease, infection, and finally death. HIV-positive individuals frequently experience undernutrition and cachexia, which are serious worldwide public health concerns that are particularly acute in sub-Saharan Africa [2].

Adults with mild to moderate acute malnutrition and severe acute malnutrition should have received therapeutic nutrition provided by a medical professional in accordance with WHO criteria [8] Malnutrition in HIV-positive people can be effectively managed with nutritional supplies such ready-to-use therapeutic meals. A plump nut or plump sup in sealed packaging is a ready-to-use therapeutic food used to treat acute malnutrition. Plumpy sup, used to treat mild to moderate acute malnutrition, and plumpy nuts, used to treat severe acute malnutrition, are both included in the peanut-based paste that also contains sugar, vegetables, fat, and skim milk [9–12].

The food prescription program (FBP) offers therapeutic or supplemental feeding, nutritional assessment, and counseling at healthcare facilities to malnourished HIV-positive adults. The program was created to supplement the ART program and fill in any gaps in HIV palliative care's nutritional intervention. This therapeutic diet is given for a predefined period of time, often for 3–6 months, depending on nutritional assessment, admission, and discharge criteria [13,14].

In order to increase the use of prescription diets, the Ethiopian Federal Ministry of Health has partnered with USAID Ethiopia, UNICEF, and the FANTA project as of 2010. In an integrated HIV and malnutrition program, 2 sachets of RUTF (Plumpy Sup) are given to an adult patient with moderate acute malnutrition (MAM) every day for a minimum of 3 months and a maximum of 4 months. A minimum of six months and a maximum of eight months will be given to patients with severe acute malnutrition who would instead get four sachets (plumpy nut) every day [14,15]. For individuals with HIV, obtaining the optimal quality of life is essential for preventing opportunistic infections, preserving immunological function, enhancing response to ART treatment, and improving immune response. Patients can postpone development to disease stages of infection, maximize the antiretroviral benefit, and enhance adherence to treatment regimens by incorporating dietary interventions into chronic HIV therapy [7].

The results of nutritional therapy with RUTF were influenced by a number of variables. Numerous studies show that immunological and clinical factors, including WHO stages, OI, CD4 count, level of adherence, and baseline nutritional status, have an impact on the nutritional treatment outcomes in patients with HIV/AIDS. These factors include age, gender, educational attainment, marital status, and occupational status [2].

Studies on malnourished HIV-positive individuals have been undertaken all over the world, and the majority of them point to sub-Saharan Africa, especially Ethiopia, as the region with the worst problem. Numerous studies have demonstrated that gender, baseline nutritional state upon admission, and a lack of frequent follow-up are significant predictors of the nutritional treatment outcome. Low recovery rates following RUTF treatment have been reported in studies from the Kenyan, Addis Ababa, and Amhara regions (40.6%, 36.3%, and 41%, respectively). According to all of these studies, the recovery rate from malnutrition following RUTF treatment is less than 50% and is still challenging [6,16,17].

Despite numerous researches on malnourished HIV-positive individuals, there was little data on the nutritional outcome and its determinants among HIV-positive adults receiving RUTF in our study area. Determining the status of nutritional outcomes and their predictors among adult HIV-positive patients receiving RUTF at west Guji Zone healthcare facilities is the purpose of the current study, which will also serve as a resource for program implementers and policymakers.

## Materials and methods

### Study area

The study was conducted in the west Guji zone, Oromia regional state southern Ethiopia, which is 467km away from Addis Ababa to the south. The Zone is surrounded in North by Gedeo Zone of SNNPs and Sidama regional state, in South by Borena Zone, in East by East

Guji Zone and in West by Burji and Amaro Special Woredas of SNNPs. The zone is formed with nine districts and two administrative towns. It has a total population of 1,424,267 of which 105,443 are urban residents. The West Guji Zone has one general hospital, two primary hospitals and 42 health centers. Among which 2 hospitals and 6 health centers provide ART services. Among them Bule Hora general hospital and Kercha primary Hospital are the largest facility with food by prescription service without interruption. Those Hospitals provides ART services for 1552 people living with HIV during study period and of these, 722 patients are on nutritional intervention programs.

## Study design and period

A facility-based retrospective cohort study was conducted at West Guji Zone public hospitals. The actual data collection (record review from patient cards and registration books) was conducted from February 1–30, 2023.

## Population of study

**Source population.** All under-nourished HIV-positive adults treated with ready-to-use therapeutic food at West Guji Zone public hospitals were the source population of the study.

**Study population.** All under-nourished HIV-infected adults registered in therapeutic feeding centers of public hospitals in West Guji Zone between January 1, 2018 and December 30, 2022 was the study population.

## Eligibility criteria

**Inclusion criteria.** All adult patients 18 years old and above diagnosed with HIV and enrolled in the therapeutic feeding program, patients with documented records of nutritional status, patients classified as undernourished based on recognized nutritional assessment tools and have complete records of demographic information, clinical history and treatment details were included.

**Exclusion criteria.** Cases with no charts available at the time of data collection were excluded. Patients with incomplete information (missing baseline data like: nutritional status, height, weight, MUAC) on food by prescription registration at the time of data collection and edematous patients were also omitted.

**Sample size determination and sampling procedure.** Based on the following assumptions, the sample size was determined using a double population proportion formula for survival analysis: 80% statistical power, a 95% degree of confidence, and the cumulative incidence rate for recovery from the earlier, comparable study carried out in Debre Markos Comprehensive Specialized Hospital [2]

n = ((Z1-α/2+ Z1-β)2)/(1og (HR)2 p*q*P(E)), where Z1-α/2 has a 95% confidence level of 1.96; Z1-β has a power of 80% (Z1-β is 0.84); the adjusted hazard rate (HR) is 11.0; P is the proportion of mild to moderately undernourished people, which is 64.7% (0.647); is the proportion of severely undernourished people, which is 35.3% (0.353); and is the incidence rate of recovery, which is 10.65% (0.1065) from similar studies, which yields a sample size of 297, and by adding the non-response rate, the sample size was 327. However, for this study, we have considered 348 patients with HIV/AIDS with treatment.

After computing the minimum adequate sample size, Adult HIV-positive patients who fulfilled inclusion criteria were selected from available list of food by prescription registration book from January 1, 2018 to December 30, 2022 (Sampling frame, 722 clients i.e., 684 clients at Bule hora General Hospital and 38 at Kercha Primary Hospital were on food by prescription therapy during study time). Then Sample was distributed proportionally to selected Hospitals

based on number of malnourished adult HIV positive patients. Accordingly, For Bule Hora General Hospital total sample was 348*684/722 = 330 and for Kercha primary Hospital sample was; 348*38/722 = 18. Finally, after recording all eligible clients, 348 participants were selected by using simple Random Sampling (Computer Generated list method by excel) in both Hospitals.

## Data collection tools and procedures

This study used secondary data that were collected by using structured data extraction checklist which was developed by reviewing different literatures and national guidelines for treatment of malnourished patients with HIV/AIDS.

By using these data extraction tool, variables such as: socio-demographic characteristics, ART related clinical condition, nutritional status, admission and discharge date from therapeutic feeding program, months on nutritional treatment and final outcomes were collected by trained data collectors from ART and food by prescription registers. Prior to data collection, the records were checked for completeness.

## Variables

The dependent variable, assessed in months, was the length of time it took for an HIV/AIDS patient who had been malnourished to recover after being admitted to West Guji Hospital's therapeutic feeding program. However, the event of interest is recovered, Recovered patients were assigned a number of 1, and censored data was assigned a number of 0. These numbers were used to represent cases in which malnourished patients with RUTF were lost to follow-up, transferred, or died prior to experiencing the event of interest, as well as those in which the event of interest had not yet occurred during the monitored Periods.

The independent variables are:—Socio demographic characteristics (age, sex, marital status, religion, educational status, Employment and residence); Clinical and immunological (ARV regimen, duration on ART, level of adherence, WHO clinical stage, opportunistic infection, Hgb, CD4 count and functional status); Base line nutritional status (SAM and MAM).

## Operational definition

**Follow-up (period of observation):** The number of months the HIV patients enrolled in the therapeutic feeding program were followed from the starting of therapy until they developed the event of interest was between January 1, 2018 and December 30, 2022.

**Censored**; Individuals who were censored were deceased patients, defaulted participants, and patients on RUTF who were not recovered at the end of the study.

**Adult**:—Age equal to or greater than 18 years old.

**Recovered:**—Patients who reached BMI of 18.5kg/m$^2$ for two consecutive visits within 3 months for MAM and within 6 months for SAM.

**Non-response;**—participants who didn't reach a BMI of 18.5kg/m$^2$ for two consecutive visits within 3 months for MAM and within 6 months for SAM.

**Defaulter**:—participants did not reach a BMI of 18.5kg/m$^2$ and drop out of the program before the end of treatment.

**Died**:—participants who died during the course of treatment and his/her death is documented in the registration book.

**Food by prescription**:—The ready to use therapeutic food prescribed to HIV-positive patients based on their nutritional status (MAM or SAM) as the strategy to address malnutrition among PLHIV through nutritional assessment, counseling and support.

**Good adherence**:—If the percentage of used dose is >95(<2 doses of 30 dose or <3 doses of 60 dose missed).

**Fair adherence**:—If percentage of used dose between 85–94 (3–5 doses of 30 dose or 4–8 doses of 60 dose missed).

**Poor adherence**:—If used dose <85(> = 6 doses of 30 dose or > = 9 doses of 60 dose missed)

## Data quality control

Prior to data collection training was given for data collectors who were trained on comprehensive HIV care, RUTF and involved in patient follow ups. The entire data collection process was closely supervised by one supervisor. During data collection process the filled checklist was checked for completeness, consistency and accuracy by principal investigator.

## Data analysis

Data was entered by using Epi data version 3.1 and exported to SPSS version 26 software for analysis. Data was cleaned and edited before analysis. Results of the study was organized and presented by using texts, figures and tables. Patients' cohort characteristics of categorical data were described in terms of frequency distribution while median, median and interquartile range, were computed for continuous data.

The Kaplan-Meier survival curve and log rank test was used to predict to the time to recovery and to compare the survival curves across baseline categorical variables. Prior to running the regression analysis, Multi-collinearity was checked. The proportional hazard assumption was checked using the schoenfeld residuals. Bivariable and multivariable cox proportional hazard regression models were fitted to identify independent predictor of nutritional recovery.

Variables having p-value < = 0.25 in bivariable analysis was fitted to multivariable analysis after checking for the assumption. Finally, p-value less than 0.05 in multivariable analysis were considered as predictor of recovery. Adjusted hazard ratio (AHR) with 95% CI was used to report strength of association and statistical significance.

## Ethical consideration

Ethical clearance was obtained from Dilla University College of Health and Medical Sciences' Ethical Review Committee (Ref no: duchm/irb/009/2023). In addition, after outlining the study's methodology and objectives, official letters were obtained from the West Guji Zone Health Department and the healthcare facilities where it was carried out. To get authorization to collect data from registration books and patient cards, the nature of the study was properly explained to the head of the department of the ART clinic. Data from the patient records that are currently in use was used in the study. To prevent a third party from accessing the data, the collected information was kept secure during the whole research project. The Dilla University College of Health Sciences Institutional Review Board (CHS-IRB) waived the informed consent requirement for this study because it was retrospective in nature. Since the study was carried out by looking over medical information, consent to participate was waived. No harm was done to any specific patients. The information should not be used for any other reasons outside of this study. Patient names and the specific ART number were not included in the data collection format in order to preserve the confidentiality of the data.

## Results

### Socio-demographic characteristics

A total of 348 with 100% response rate of research participants participated in this study. Almost half of the participants were females (53.4%) and the Median ages of participants were 38 years (IQR 32–49). In this study, 46.8% of her participants had no formal education and

**Table 1. Socio-demographic characteristics of malnourished adult HIV-positive patients at admission by outcome in West Guji Zone public hospitals, Southern Ethiopia, 2023.**

| Participant characteristics | Variables | Graduated/ recovered | Defaulter | Nonresponse | Death | Total |
|---|---|---|---|---|---|---|
| **Sex** | Male | 73(21%) | 40(11.5%) | 47(13.5%) | 2(0.6%) | 162(46.6%) |
| | Female | 125(35.9%) | 21(6%) | 37(10.6%) | 3(0.9%) | 186(53.4%) |
| **Age in years Categorized (18–71)** | 18–29 | 55(15.8%) | 9(2.6%) | 8(2.3%) | 0(0.0%) | 72(20.7%) |
| | 30–39 | 83(23.9%) | 21(6%) | 10(2.9%) | 3(0.9%) | 117(33.6%) |
| | 40–49 | 30(8.6%) | 25(7.2%) | 22(6.3%) | 0(0.0%) | 77(22.1%) |
| | > = 50 | 30(8.6%) | 6(1.7%) | 44(12.6%) | 2(0.6%) | 82(23.6%) |
| **Educational Status** | No formal education | 72(44.2%) | 23(14.1%) | 63(38.7%) | 5(3.1%) | 163(46.8%) |
| | Primary | 52(63.4%) | 25(30.5%) | 5(6.1%) | 0(0.0%) | 82(23.6%) |
| | Secondary | 60(67.4%) | 13(14.6%) | 16(18%) | 0(0.0%) | 89(25.6%) |
| | Higher education | 14(100%) | 0(0.0%) | 0(0.0%) | 0(0.0%) | 14(4%) |
| **Employment status** | Self employed | 81(67.5%) | 13(10.8%) | 26(21.7%) | 0(0.0%) | 120(34.5%) |
| | Government/ NGO employee | 10(100%) | 0(0.0%) | 0(0.0%) | 0(0.0%) | 10(2.9%) |
| | Unemployed | 107(49.1%) | 48(22%) | 58(26.6%) | 5(2.3%) | 218(62.6%) |
| **Marital status** | Single | 32(61.5%) | 16(30.8%) | 4(7.7%) | 0(0.0%) | 52(14.9%) |
| | Married | 142(56.8%) | 40(16%) | 66(26.4%) | 2(0.8%) | 250(71.8%) |
| | Divorced | 4(18.2%) | 5(22.7%) | 10(45.5%) | 3(13.6%) | 22(6.3%) |
| | Widowed | 20(83.3%) | 0(0.0%) | 4(16.7%) | 0(0.0%) | 24(6.9%) |
| **Residence** | Rural | 80(51.6%) | 10(6.5%) | 60(38.7%) | 5(3.2%) | 155(44.5%) |
| | Urban | 118(61.1%) | 51(26.4%) | 24(12.4%) | 0(0.0%) | 193(55.5%) |
| **Religion** | Orthodox | 48(75%) | 8(12.5%) | 8(12.5%) | 0(0.0%) | 64(18.4%) |
| | Protestant | 101(58.4%) | 38(22%) | 34(19.7%) | 0(0.0%) | 173(49.7%) |
| | Muslim | 41(45.6%) | 14(15.6%) | 32(35.6%) | 3(3.3%) | 90(25.9%) |
| | Other | 8(38.1%) | 1(4.8%) | 10(47.6%) | 2(9.5%) | 21(6%) |

only 14 (4%) had a higher level of education. More than half of study participants (62.6%) were unemployed and the rest (34.5%) were self-employed. More than two-thirds of the study participants were married (71.8%), while 6.3% were divorced. More than half of the participants (55.5%) live in urban areas. Nearly half of the participants (49.7%) were followers of protestant Christianity (Table 1).

## Clinical characteristics of HIV-infected adult patients at admission

From a total of 348 study participants, more than three-fourths were on WHO clinical stages I and II (77.6%), and 91.1% were on a first-line ART regimen. Nearly 75% of study participants had good ART drug adherence. More than half of study participants (58.6%) had no baseline opportunistic infection during admission. Nearly two-thirds of study participants were not taking cotrimoxazole preventive therapy at admission, while 81.3% were taking isoniazid preventive therapy. 125 (35.9%) of study participants had CD4 levels below threshold (< 200 cells/m3). Almost all (98.6%) of study participants were on HAART at admission. 98 (28.2%) of the study participants were treated as SAM cases, while the rest were moderately malnourished. The median BMI was 16.80 kg/m$^2$ (IQR 15.4–16.9) at admission and 18.8 kg/m$^2$ (IQR 17.8–19.6) at the end of the study (Table 2).

## Characteristics of recovered patients from malnutrition

Study participants had minimum of 1 month and maximum follow-up period of 10 months with median follow-up of 3 months (IQR 3–5 months), during which time 198 (56.9%) HIV-

**Table 2. Baseline clinical characteristics of adult malnourished HIV patients enrolled in the therapeutic feeding program, West Guji Zone public hospitals, Southern Ethiopia, 2023.**

| Variables | Category | Treatment outcome N (%) | | | | Total(n = 348) |
|---|---|---|---|---|---|---|
| | | Recovered | Defaulter | Nonresponse | Death | |
| WHO clinical stage | Stage I & II | 174(64.4) | 51(18.9) | 45(16.7%) | 0(0.0) | 270(77.6) |
| | Stage III & IV | 24(30.8) | 10(12.8) | 39(50.0) | 5(6.4) | 78(22.4) |
| ARV regimen | First line | 198(62.5) | 56(17.7) | 60(18.9) | 3(0.9) | 317(91.1) |
| | Second line | 0(0.0) | 5(16.1) | 24(77.4) | 2(6.5) | 31(8.9) |
| ART adherence | Good | 165(63.5) | 47(18.1) | 48(18.5) | 0(0.0) | 260(74.7) |
| | Fair | 33(63.5) | 9(17.3) | 7(13.5) | 3(5.8) | 52(14.9) |
| | Poor | 0(0.0) | 5(13.9) | 29(80.6) | 2(5.6) | 36(10.4) |
| Baseline OIs | Yes | 75(52.1) | 18(12.5) | 46(31.9) | 5(3.5) | 144(41.4) |
| | No | 123(60.3) | 43(21.1) | 38(18.6) | 0(0.0) | 204(58.6) |
| CPT status | Yes | 50(41.7) | 22(18.3) | 43(35.8) | 5(4.2) | 120(34.5) |
| | No | 148(64.9) | 39(17.1) | 39(17.1) | 0(0.0) | 228(65.5) |
| INH Status | Yes | 157(55.5) | 47(16.6) | 74(26.1) | 5(1.8) | 283(81.3) |
| | No | 41(63.1) | 14(21.5) | 10(15.4) | 0(0.0) | 65(18.7) |
| Other medication | Yes | 63(42.3) | 26(17.4) | 55(36.9) | 5(3.4) | 149(42.8) |
| | No | 135(67.8) | 35(17.6) | 29(14.6) | 0(0.0) | 199(57.2) |
| Baseline CD4 level | <200cells/m$^3$ | 53(42.4) | 13(10.4) | 54(43.2) | 5(4.0) | 125(35.9) |
| | 200-350cells/m$^3$ | 125(61.6) | 48(23.6) | 30(14.8) | 0(0.0) | 203(58.3) |
| | >350cells/m$^3$ | 20(100.0) | 0(0.0) | 0(0.0) | 0(0.0) | 20(5.7) |
| Hgb level | <10g/dl | 5(55.6) | 0(0.0) | 1(11.1) | 3(33.3) | 9(2.6) |
| | 10–11.99 g/dl | 148(51.7) | 57(19.9) | 79(27.6) | 2(0.7) | 286(82.2) |
| | > = 12g/dl | 45(84.9) | 4(7.5) | 4(7.5) | 0(0.0) | 53(15.2) |
| Functional status | Working | 146(72.6) | 34(16.9) | 21(10.4) | 090.0) | 201(57.8) |
| | Ambulatory | 52(43.7) | 26(21.8) | 38(31.9) | 3(2.5) | 119(34.2) |
| | Bed ridden | 0(0.0) | 1(3.6) | 25(89.3) | 2(7.1) | 28(8.0) |
| ART status | On HAART | 193(56.3) | 61(17.8) | 84(24.5) | 5(1.5) | 343(98.6) |
| | Pre ART | 5(100.0) | 0(0.0) | 0(0.0) | 0(0.0) | 5(1.4) |
| Months on ART | <6 months | 94(55.3) | 25(14.7) | 48(28.2) | 3(1.8) | 170(48.9) |
| | 6–12 months | 69(71.9) | 23(24.0) | 4(4.2) | 0(0.0) | 96(27.6) |
| | 12 months and above | 35(42.7) | 13(15.9) | 32(39.0) | 2(2.4) | 82(23.6) |
| Baseline nutritional status | SAM | 23(23.5) | 11(11.2) | 59(60.2) | 5(5.1) | 98(28.2) |
| | MAM | 175(70.0) | 50(20.0) | 25(10.0) | 0(0.0) | 250(71.8) |

positive adults recovered from malnutrition after receiving prescription nutrition therapy. Of the 348 study participants, 150 (43.1%) had poor nutritional outcome; of these, 61 (17.5%) were defaulter, 84 (24.1%) did not respond to the prescribed food therapy and the rest (5 (1.4%)) died during nutritional therapy (Fig 1).Generally, it was discovered that the recovery rate per 100 person-month observation was 9.83(95% CI; 3.12, 13.44).

## Kaplan-Meier survival estimate recovery time from under nutrition

The overall median recovery time was 5.34 months (95% CI: 5.02, 5.74 months) (Fig 2).

In terms of estimated recovery time by Age Category, Patients in Age between 18–29 had an estimated median recovery time of 3.30 months (95% CI: 3.13, 3.48 months), Age between 30–39 had 4.18 months (95% CI: 3.74, 4.62 months) and age between 40–49 had estimated recovery time of 5.89 months (95%CI: 4.95, 6.82 months) when Compared to patients with age > = 50 years which had estimated recovery time of 7.61 months (95% CI:6.92, 8.30 months) (Fig 3).

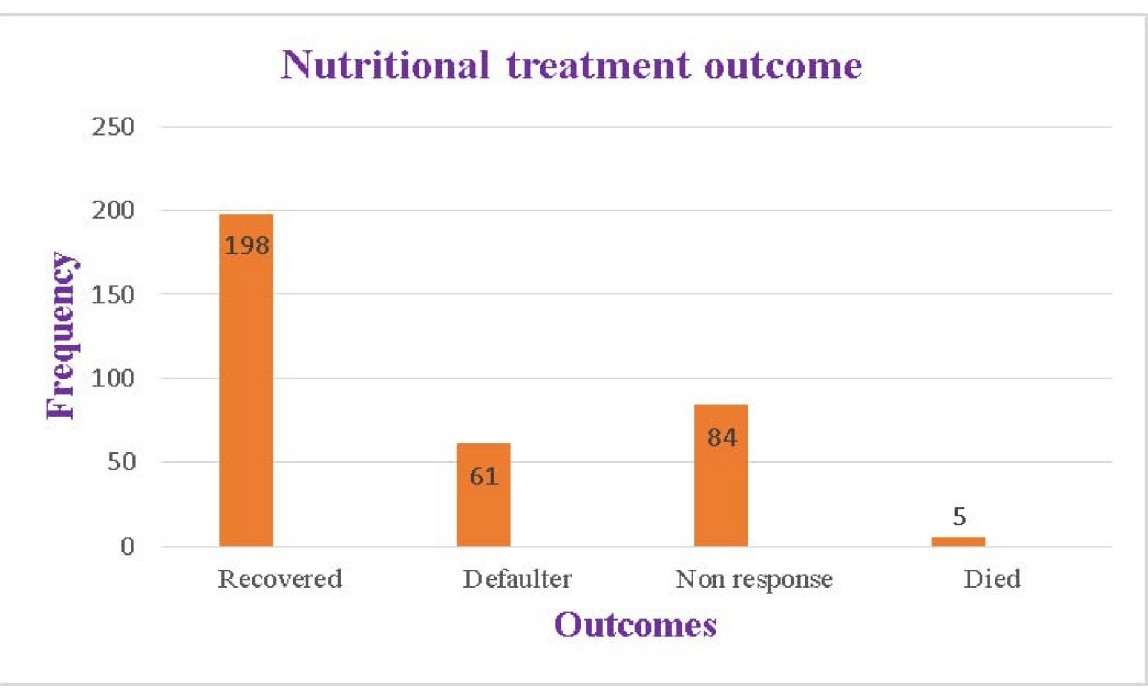

**Fig 1. Nutritional treatment outcomes of malnourished adult HIV-positive patients at West Guji Zone public hospitals, Southern Ethiopia, 2023.**

## Factors associated with recovery as treatment outcomes

By using bivariate Cox regression analysis, variables with a p-value of < 0.25 were recruited to be included in the final model. Thus, in multivariable analysis, the variables found to be associated with recovery from under nutrition from food by prescription in the final model are marital status, WHO clinical stage, age, and functional status. Therefore, this study found that divorced adult HIV-positive patients had a 79% lower probability of recovery from malnutrition when compared to married ones (AHR = 0.21; 95% CI: 0.06, 0.69).

Adult HIV positive patients on WHO clinical stages III and IV had a 58% lower probability of recovery from malnutrition after being enrolled in food by prescription therapy compared with those in WHO clinical stages I and II (AHR = 0.42; 95% CI; 0.23, 0.79). Study participants whose ages were 18–29 years old (AHR = 2.39; 95% CI: 1.33, 4.32) or 30–39 years old (AHR = 2.64; CI: 1.56, 4.46) had a higher probability of recovering from malnutrition when compared to participants over the age of 50. Moreover, adult HIV positive patients who were enrolled in food by prescription therapy with working functional status had a two-times higher probability of recovery from malnutrition when compared to those who were ambulatory and bedridden (AHR = 2.00; 95% CI: 1.25, 3.23). Having opportunistic infections, CD4 levels below threshold, and any stage of educational level or residence were unrelated to the probability of nutritional recovery (Table 3).

## Factors associated with defaulting from under nutrition in food by prescription therapy

During Bivariate analyses, 6 variables were found to be candidates for Multivariable analysis with P-value of < 0.25. These variables include Residence, WHO Clinical stage, OIs, Base line nutritional status, age and functional status. After performing multivariate analyses only

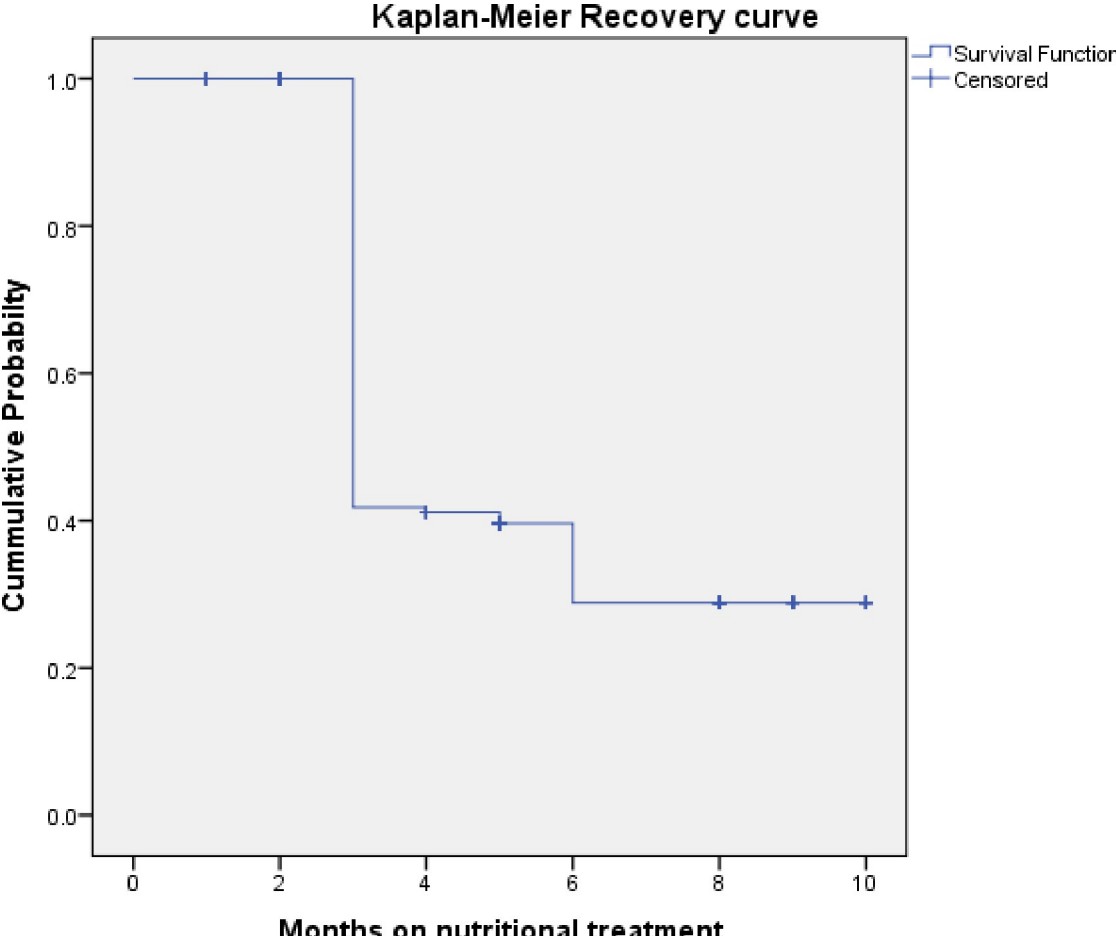

**Fig 2. Shows overall Kaplan-Meier estimation of survival time to recover from under nutrition among adult HIV positive patients at West Guji Zone public health care facilities, Southern Ethiopia, from January 01, 2018 to December 30, 2022.**

WHO clinical stage was significant (AHR = 8.39; 95% CI: 1.48, 47.50), with P-value of 0.016 (Table 4).

## Factors associated with nonresponse to treatment with food by prescription

During Bivariate analyses, 7 variables were found to be candidates for multivariable analyses with P-value <0.25. These variables were Residence, WHO clinical stage, ARV regimen, OIs, patients on other medication, low base line CD4 and functional status. After multivariate analyses only Functional status was found to be significant (AHR = 2.76; 95% CI = 1.18, 6.47), P-value = 0.019. (Table 5)

## Discussion

Severe malnutrition was a typical ailment among HIV patients. Prescription food treatment has been used as a regular intervention to treat this problem and prevent the debilitating wasting condition that is linked to persistent infection. It is thought that giving patients additional nutrients for a while can enhance their ability to gain weight and function [18].

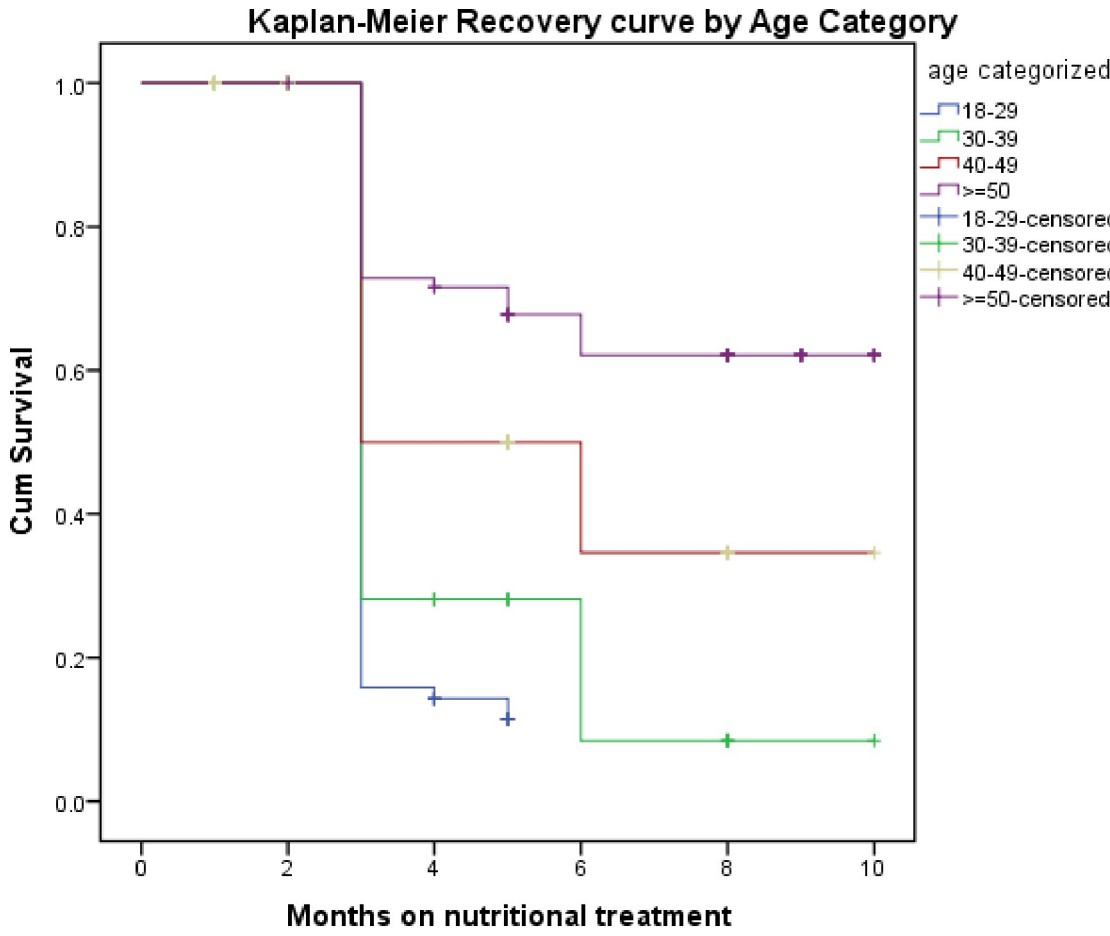

**Fig 3. The estimated Kaplan-Meier recovery curves of under-nourished HIV positive adults treated with RUTF by age category at West Guji Zone healthcare facilities from January 01, 2018 to December 30, 2022.**

This current study shows that more than half (56.9%) of adult HIV-positive patients recovered from under nutrition. This is similar to a study done in the Tigray region of Ethiopia, where 55.3% of study participants recovered from malnutrition. In that study, 56.9% of adult HIV-positive patients who were enrolled in food therapy on prescription also recovered from under nutrition [19], but it's less than research from southwest Ethiopia [20]. These variations might be explained by the various study settings and timespan.

According to percentage terms, the recovery rate in this study is higher than the 35.3% recorded in a study conducted at Gondar University Hospital in Northwest Ethiopia. Since the current study only focuses on malnourished HIV/AIDS-infected adults, it is possible that the lower recovery rate reported from Gondar University Hospital is due to the study group consisting of both adults and children. While only 68.1 percent of patients on ART were participating in RUTF, according to a Gondar study, our study's better recovery rate may also have been due to the fact that all participants got integrated ART and RUTF therapy [21]. The median recovery length for nutritional recovery in this study was 5.34 months (95% CI: 5.02, 5.74 months). This result is consistent with a study conducted at Debre Markos Comprehensive Specialized Hospital, which discovered a 5-month median recovery period (IQR: 3, 8) [2]. A further SSA study found that the average recovery period was 3.3 months [5].

**Table 3. Bi-variable and multi-variable cox regression analysis of predictors of recovery from under-nutrition among patients on therapeutic feeding program at West Guji Zone public hospitals, Southern Ethiopia, 2023.**

| Variables | Category | CHR (95% CI) | AHR (95% CI) | P-value |
|---|---|---|---|---|
| **Sex** | Male | 1.00 | 1.00 | |
| | Female | 0.69(0.52, 0.92) | 0.84(0.60, 1.19) | 0.329 |
| **Educational Status** | No education | 0.40(0.22, 0.71) | 0.94(0.48, 1.84) | 0.865 |
| | Primary | 0.81(0.45, 1.46) | 0.86(0.43, 1.72) | 0.665 |
| | Secondary | 0.69(0.37, 1.24) | 0.56(0.27, 1.15) | 0.113 |
| | Higher Education | 1.00 | 1.00 | |
| **Marital Status** | Single | 1.00 | 1.00 | |
| | Married | 0.68(0.46, o.99) | 0.81(0.48, 1.35) | 0.414 |
| | Divorced | 0.24(0.08, 0.67) | 0.21(0.06, 0.69) | 0.010 |
| | Widowed | 0.93(0.53, 1.62) | 0.84(0.38, 1.89) | 0.680 |
| **Residence** | Rural | 1.81(1.36, 2.41) | 0.85(0.54, 1.33) | 0.471 |
| | Urban | 1.00 | 1.00 | |
| **WHO clinical stage** | Stage I & II | 1.00 | 1.00 | |
| | Stage III & IV | 0.32(0.21, 0.49) | 0.42(0.23, 0.79) | 0.007 |
| **OIs** | Yes | 1.00 | 1.00 | |
| | No | 0.68(0.51, 0.90) | 1.28(0.87, 1.88) | 0.205 |
| **Base line CD4 status** | $<200$cells/m$^3$ | 1.00 | 1.00 | |
| | 200-350cells/m$^3$ | 0.36(0.21, 0.61) | 1.48(0.97, 2.25) | 0.071 |
| | $>350$cells/m$^3$ | 0.69(0.43, 1.04) | 1.43(0.70, 2.92) | 0.324 |
| **Functional status** | Working | 0.36(0.26, 0.50) | 2.00(1.25, 3.23) | 0.004 |
| | Ambulatory and bed ridden | 1.00 | 1.00 | |
| **Age Categorized (18–71)** | 18–29 | 3.45(2.17, 5.48) | 2.39(1.33, 4.32) | 0.004 |
| | 30–39 | 2.95(1.93, 4.51) | 2.64(1.56, 4.46) | $<0.001$ |
| | 40–49 | 1.88(1.13, 3.13) | 1.45 (0.76, 2.75) | 0.255 |
| | $>50$ | 1.00 | 1.00 | |

**Table 4. Bi-variable and multi-variable cox regression analysis of predictors of defaulting from food by prescription treatment programs among patients on therapeutic feeding program at West Guji Zone public hospitals, Southern Ethiopia, 2023.**

| Variables | Category | CHR(95% CI) | AHR (95% CI) | P-Value |
|---|---|---|---|---|
| **Residence** | Rural | 1.00 | 1.00 | |
| | Urban | 0.38(0.15,0.95) | 0.51(0.16, 1.58) | 0.243 |
| **WHO Clinical Stage** | Stage I & II | 1.00 | 1.00 | |
| | Stage III & IV | 2.65(1.06, 6.64) | 8.39(1.48, 47.5) | 0.016 |
| **OIs** | Yes | 1.00 | 1.00 | |
| | No | 0.68(0.37, 1.27) | 3.03(0.91, 10.13) | 0.072 |
| **Nutritional Status** | SAM | 1.000 | 1.000 | |
| | MAM | 0.41(0.18, 0.96) | 3.03(0.51, 18.05) | 0.224 |
| **Age in years** | 18–29 | 6(0.76, 47.36) | 1.10(0.66,19.21) | 0.946 |
| | 30–39 | 7.06(0.95, 52.47) | 5.13(0.18, 48.67) | 0.341 |
| | 40–49 | 9.64(1.30, 71.22) | 9.26(0.41,51.56) | 0.163 |
| | $>= 50$ | 1.00 | 1.00 | |
| **Functional Status** | Working | 1.00 | 1.00 | |
| | Ambulatory and bed ridden | 1.50(0.87, 2.56) | 0.71(0.32, 1.60) | 0.406 |

**Table 5. Bi-variable and multi-variable cox regression analysis of predictors of non-respond to food by prescription therapy among patients on therapeutic feeding program at West Guji Zone public hospitals, Southern Ethiopia, 2023.**

| Variables | Category | CHR(95% CI) | AHR(95% CI) | P-Value |
|---|---|---|---|---|
| Residence | Rural | 1.00 | 1.00 | |
| | Urban | 0.52(0.31, 0.87) | 0.94(0.477, 1.71) | 0.756 |
| WHO Clinical Stage | Stage I & II | 1.00 | 1.00 | |
| | Stage III & IV | 1.68(1.09, 2.61) | 0.71(0.32, 1.56) | 0.395 |
| ARV Regimen | First Line | 1.00 | 1.00 | |
| | Second Line | 1.98(1.21, 3.24) | 1.78(0.89, 3.57) | 0.105 |
| OIs | Yes | 1.00 | 1.00 | |
| | No | 0.55(0.35, 2.82) | 0.91(0.48, 1.73) | 0.771 |
| On Other Medication | Yes | 1.00 | 1.00 | |
| | No | 0.39(0.24, 0.64) | 0.48(0.08, 2.84) | 0.419 |
| Base line CD4 level | < 200 Cells/m$^3$ | 1.00 | 1.00 | |
| | 200–350 Cells/m$^3$ | 0.53(0.33, 0.87) | 1.72(0.42, 6.95) | 0.450 |
| Functional status | Working | 1.00 | 1.00 | |
| | Ambulatory and Bedridden | 3.71(2.14, 6.42) | 2.76(1.18, 6.47) | 0.019 |

This study found factors that predicted the participants' time to recover from malnutrition which were marital status, WHO clinical stage, age, and functional level. Marital status was one of the socio-demographic factors linked to nutritional treatment recovery.

Divorced participants had a 79% lower chance of recovering from malnutrition after enrolling in food by prescription therapy as compared to single participants. This conclusion might be explained by the fact that persons who are single are less concerned with caring for dependents or other financial or social issues, which may help with nutritional recuperation. This study's findings are consistent with those of studies conducted in south-western Ethiopia [20].

Age of the participants was significantly related with nutritional treatment recovery. When compared to patients over the age of 50, patients ages 18 to 29 had 2.4 and 30–39 had 2.6 times higher chances of recovering from malnutrition, respectively. The results of this study are at odds with those from Uganda and Kenya [5]. In our study, more than half of the patients (54.3%) were aged between 18 and 39. This ratio may have contributed to the disparity in nutritional recovery. Additionally, almost 80% of the patients in this age group had WHO clinical stages I or II, indicating a strong likelihood of recovery.

WHO clinical stage was one of the factors that were significantly correlated with nutritional recovery after treatment with food by prescription therapy. The likelihood of individuals with advanced WHO clinical stages III and IV recovering from malnutrition was reduced by 58%. This could be the result of patients receiving insufficient nutritional treatment as a result of severe illness, which weakens the immune system and lowers nutritional intake. This finding of study is supported by evidence from retrospective cohort study conducted at Finote Selam hospital Northern Ethiopia.

Another factor related to nutritional treatment recovery was the functional state of the patients participating in the therapeutic feeding program. Adult HIV-positive persons with working functional status had twice the chance of recovery compared to bedridden and ambulatory patients. This might be explained by the fact that patients who are bedridden and ambulatory can't take RUTF on time, and that having many conditions may make it more likely that they won't get enough nourishment. This outcome is in line with the Debre Markos Comprehensive Specialty Hospital's findings [2]. According to this study, 17.5% of study subjects dropped out of the malnutrition treatment program. This result is lower than that of a survey

done in Kenya and Uganda, where 22.6% of respondents were defaulters [5]. This discrepancy may be the result of the study's utilization of a bigger sample size and various patient follow-up methods.

This study revealed a relationship between stopping nutritional treatment programs and the WHO clinical stage at baseline. The chance of discontinuing treatment was 8.4 times higher for patient in advanced WHO clinical stages (II and IV) than for those in stages I and II. The reason behind this could be that the majority of people who experience severe illness also didn't complete the ART treatment plan and didn't visit hospitals on time. Retrospective cohort study data from the Tigray region of northern Ethiopia supports this finding [19]. This study also shows that food-based prescription therapy did not respond for 24.1% of study participants. Patients with ambulatory and bedridden functional status showed a 2.76 times higher risk of therapy non-response compared to those with a functioning functional status. Patients who are ambulatory yet bedridden may not visit hospitals or receive nutritional therapy on time, which might lead to this. Mekele Hospital data supports this conclusion [22].

### Limitation of study

Due to the retrospective nature of this study's design, the analyses of the factors that influence nutritional outcome were restricted to the information that was only recorded in patient files. As a result, information on some socio-demographic traits that were unavailable in patient records, such as household food security, economic status, and other clinically relevant aspects, was left out. The absence of a control group for comparison and the possibility of data recruitment bias resulted from the samples being restricted to only people with recorded nutritional results.

### Conclusion and recommendation

In this study, out of 100 undernourished HIV patients who were participating in RUTF followed up for a month, 10 of them had a recovered. Being divorced and having WHO clinical stages III and IV at admission was associated with negative predictors of nutritional recovery while, having working functional status, and being between the ages of 18 and 39 were positive predictors. Advanced WHO clinical stages (III and IV) are associated with an increased risk of dropping out of a nutritional treatment program, whereas bedridden and ambulatory functional statuses are associated with an increased risk of treatment non-responsiveness.

Based on the study's findings, we recommend ART program managers, physicians, and health authorities to place a strong emphasis on treatment by food on patients with advanced WHO clinical stages, bedridden and ambulatory functional status, those who are divorced, and those who are older than 50 years old. Finally, we suggest that future researchers do multicenter studies in order to gather more data by taking into account additional parameters that were not considered in this study.

### Supporting information

**S1 File. Questionar.**
(PDF)

### Acknowledgments

We would like to thank Dilla University College of Health and Medical Science, School of Public health, Reproductive Health Department for giving us this opportunity and for the overall support provided to undertake this thesis. We would also like to thank West Guji Zone Health

Department, Bule Hora General and Kercha Primary Hospital for giving permission and their kind cooperation during data collection. Finally, we would like to thank all data collectors and supervisors.

## Author Contributions

**Conceptualization:** Eden Ashenafi.

**Data curation:** Eden Ashenafi, Getahun Beyene Guluma.

**Formal analysis:** Eden Ashenafi, Dirshaye Argaw, Habtamu Endashaw Hareru.

**Funding acquisition:** Eden Ashenafi.

**Investigation:** Eden Ashenafi, Getahun Beyene Guluma.

**Methodology:** Eden Ashenafi, Habtamu Endashaw Hareru.

**Project administration:** Eden Ashenafi.

**Resources:** Eden Ashenafi.

**Software:** Eden Ashenafi, Habtamu Endashaw Hareru.

**Supervision:** Dirshaye Argaw, Nagasa Eshete Soboksa.

**Validation:** Eden Ashenafi, Getahun Beyene Guluma, Dirshaye Argaw, Nagasa Eshete Soboksa.

**Visualization:** Eden Ashenafi, Nagasa Eshete Soboksa.

**Writing – original draft:** Eden Ashenafi, Getahun Beyene Guluma, Dirshaye Argaw, Habtamu Endashaw Hareru, Nagasa Eshete Soboksa.

**Writing – review & editing:** Eden Ashenafi, Getahun Beyene Guluma, Dirshaye Argaw, Habtamu Endashaw Hareru, Nagasa Eshete Soboksa.

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
