## [Decision Letter · Decision Letter 0]

31 Oct 2023

PONE-D-23-28673Nutritional outcomes of Therapeutic feeding program and its predictors among undernourished adult HIV positive patients at healthcare facilities of West Guji Zone, Southern Ethiopia: A Retrospective Cohort StudyPLOS ONE

Dear Dr. Ashenafi,

Thank you for submitting your manuscript to PLOS ONE. After careful consideration, we feel that it has merit but does not fully meet PLOS ONE’s publication criteria as it currently stands. Therefore, we invite you to submit a revised version of the manuscript that addresses the points raised during the review process.

We look forward to receiving your revised manuscript.

Kind regards,

Mohammed Hasen Badeso, MPH in Field Epidemiology

Academic Editor

PLOS ONE

Journal Requirements:

4. Please ensure that you refer to Figure 1 and 2 in your text as, if accepted, production will need this reference to link the reader to the figure.

5. Please upload a copy of Figure 1 and 2, to which you refer in your text on page 12. If the figure is no longer to be included as part of the submission please remove all reference to it within the text.

6. We note you have included a table to which you do not refer in the text of your manuscript. Please ensure that you refer to Table 1, 2 and 3 in your text; if accepted, production will need this reference to link the reader to the Table.

Reviewers' comments:

Reviewer's Responses to Questions

**Comments to the Author**

1. Is the manuscript technically sound, and do the data support the conclusions?

Reviewer #1: Yes

Reviewer #2: No

2. Has the statistical analysis been performed appropriately and rigorously? 

Reviewer #1: Yes

Reviewer #2: No

3. Have the authors made all data underlying the findings in their manuscript fully available?

Reviewer #1: Yes

Reviewer #2: Yes

4. Is the manuscript presented in an intelligible fashion and written in standard English?

Reviewer #1: No

Reviewer #2: No

5. Review Comments to the Author

Reviewer #1: Review report

Thank you very much for the opportunity to review this manuscript that addresses an important clinical subject in its field. However, I feel that there are number issues that need to address as detailed in the attached document.

Comments

• General comments

- The paper is well written but the language, grammar and punctuation needs to be improved significantly.

• Background

- This section is uninformative because it does not review the literature on the main issue of the study. The authors mainly focused on the prevalence of HIV particularly, in the first paragraphs, while the main issue is nutritional recovery among HIV/AIDS patients,

• Highlight on the epidemiological burden of undernutrition among people living with HIV/ AIDS in terms of its prevalence / rate and the relationship between the two in the background section.

Results

o Figure 1 is not self-explanatory. The X & Y axes are not labeled.

• Discussion

o There are same vague statements in the discussion section. For example, the statement that says “Age was a further socio-demographic variable connected to nutritional treatment recovery”. This statement needs to be rephrased.

o Researchers should write the implications of the main findings of their study the bridge the gap between policy and research. Better to include the implications of the major findings.

• Conclusion

o Line 469: The statement that says Nutritional recovery in this study fell short of WHO Sphere requirements. I think you mean falls short of …

Reviewer #2: The manuscript entitled “Nutritional Outcomes of Therapeutic feeding program and its predictors among undernourished adult HIV positive patients at healthcare facilities of West Guji Zone, Southern Ethiopia: A Retrospective Cohort Study” would have important contribution for decision-makers by suggesting strategies for improving the management of HIV patients, which is a significant cause of morbidity and mortality in Ethiopia. But the manuscript is not fit for publication in its current form.

General comment:

1. The manuscript has many editorial problems and needs language editing.

Abstract:

1. In the background section of the abstract, the problem under study is not well stated. I suggest rewriting it to show what the problem or gap is and why it needs to be studied.

2. The abstract is lengthy and can be made shorter by minimizing unnecessary details in the methods section.

3. The conclusion does not align with the results. For instance, you have conclude that divorced marital status was a negative predictor but you didn’t mention it in the result section.

Introduction:

1. Line 58. From the sentence “The human immunodeficiency virus and acquired immune deficiency syndrome……”, each word should be capitalized and put the abbreviation in bracket

2. From Line 67 to 68, the authors mentioned that HIV/AIDS affects 67% of people worldwide. I think this figure is exaggerated and I doubt on the credibility of your source. Check it again.

3. Abbreviations should be written in full words in their first use.

4. There is a problem of coherence. The authors discuss about the treatments of malnutrition first (line 72 to 97), and then they discuss about the magnitude and effect of malnutrition (line 99 to 107). I suggest to discuss the magnitude and effect of malnutrition among HIV patients first.

5. I believe that the problem under study (undernutrition among HIV) is not well stated. Rather the authors focus on the magnitude of HIV and undernutrition treatment procedure.

Methods:

1. The study period for a follow-up study should be the time between the initiations of follow up and end of follow up. The authors mentioned the time when they access the patients chart.

2. The study populations of this study were “All under-nourished HIV-infected adults registered in therapeutic feeding centers of public hospitals in West Guji Zone between January 1, 2018 and December 30, 2022” and the data were collected from February 1-30, 2023. But in order to assess whether the participant is recovered or not it needs a minimum of 3 months after initiation of therapeutic feeding based your operational definition. So, there should be at least a 3 months gab between the last date of subject recruitment and outcome assessment. What is your explanation here?

3. In the inclusion criteria, why do the authors need to know the nutritional treatment outcomes in order to include the participants in the study? There could be selection bias if we included participants after knowing their outcome status.

4. The authors didn’t calculate the sample size using appropriate formula for survival analysis.

5. I think the data collectors were health care providers working in the ART clinic. This could introduce observer bias, so how do you manage it?

6. The authors stated that the dependent variable was “nutritional treatment outcome (recovered or unrecovered)”, means it is a binary outcome. So why you conduct survival analysis? If you want conduct a survival analysis, you need to have a time to event data. Please make it clear.

7. The authors operationalized adherence. But is it adherence to ART or Adherence to food by prescriptions? I think both are important

8. The authors need to define the time to event and censor if they want to stick with survival analysis. They didn’t mentioned when they start and stop following the participants.

Results:

1. The minimum and maximum follow up period, median follow up period, and the total person time contribution of the study participants were not mentioned.

2. Make tables up to the standard by reducing lines

3. When I read the result section, I am confused about what the event is. The authors try fit 3 different models considering 3 events, recovery, non-response and default. You have to stick with one of it. Which one is the problem that you are interested in? Then, the introduction and the methods will be also related with the event of interest.

Discussion:

1. Start by summarizing your finding, for instance line 404 you just start with comparison

2. The discussion is poorly written which lacks adequate comparison and justifications.

3. From line 464 to 465, the authors mentioned the following as a limitation. “The absence of a control group for comparison and the possibility of data recruitment bias resulted from the samples being restricted to only people with recorded nutritional results.” How could this limitations affect the study? What kind of control group you could use?

Conclusion:

1. From line 469 t0 471, the authors conclude that “The likelihood of nutritional recovery is increased by divorce, WHO clinical stages I and II at admission, working functional status, and age between 18 and 39.” But the result doesn’t support that divorce increases the likelihood of recovery.

6. PLOS authors have the option to publish the peer review history of their article (what does this mean?). If published, this will include your full peer review and any attached files.

Reviewer #1: **Yes: **Dr. Dereje Tsegaye

Reviewer #2: No

---

## [Author Response · Author response to Decision Letter 0]

3 Jan 2024

Let me begin by thanking you for the reviewer's comments.We greatly appreciate your support and feedback, which we have tried to include in our work. we tried to the best and uploading the response to reviewer letter, which includes a detailed response to every concern raised by the reviewers.

---

## [Editor Report · Decision Letter 1]

4 Jan 2024

Nutritional outcomes of Therapeutic feeding program and its predictors among undernourished adult HIV positive patients at healthcare facilities of West Guji Zone, Southern Ethiopia: A Retrospective Cohort Study

PONE-D-23-28673R1

Dear Eden,

We’re pleased to inform you that your manuscript has been judged scientifically suitable for publication and will be formally accepted for publication once it meets all outstanding technical requirements.

Kind regards,

Mohammed Hasen Badeso, MPH in Field Epidemiology

Academic Editor

PLOS ONE
---

## [Editor Report · Acceptance letter]

15 Jan 2024

PONE-D-23-28673R1 

PLOS ONE

Dear Dr. Ashenafi, 

I'm pleased to inform you that your manuscript has been deemed suitable for publication in PLOS ONE. Congratulations! Your manuscript is now being handed over to our production team.

Kind regards, 

on behalf of

Mr Mohammed Hasen Badeso 

Academic Editor

PLOS ONE